# The anti-platelet drug cilostazol enhances heart rate and interrenal steroidogenesis and exerts a scant effect on innate immune responses in zebrafish

**Wei-Chun Chang[1,2], Mei-Jen Chen[1], Chung-Der Hsiao[3], Rong-Ze Hu[1], Yu-Shan Huang[1], Yu-Fu Chen**[1], **Tsai-Hua Yang[1], Guan-Yi Tsai[1], Chih-Wei Chou[1], Ren-Shiang Chen[1], Yung-Jen Chuang[4,5], Yi-Wen Liu**[1] *

**1** Department of Life Science, Tunghai University, Taichung, Taiwan, **2** Feng Yuan Hospital of the Ministry of Health and Welfare, Taichung, Taiwan, **3** Department of Bioscience Technology, Chung Yuan Christian University, Chung-Li, Taiwan, **4** Institute of Bioinformatics and Structural Biology, National Tsing Hua University, Hsinchu, Taiwan, **5** Department of Medical Science, National Tsing Hua University, Hsinchu, Taiwan

* dlslys@thu.edu.tw

**Data Availability Statement:** All relevant data are within the paper.

## Abstract

### Rationale

Cilostazol, an anti-platelet phosphodiesterase-3 inhibitor used for the treatment of intermittent claudication, is known for its pleiotropic effects on platelets, endothelial cells and smooth muscle cells. However, how cilostazol impacts the endocrine system and the injury-induced inflammatory processes remains unclear.

### Methods

We used the zebrafish, a simple transparent model that demonstrates rapid development and a strong regenerative ability, to test whether cilostazol influences heart rate, steroidogenesis, and the temporal and dosage effects of cilostazol on innate immune cells during tissue damage and repair.

### Results

While dosages of cilostazol from 10 to 100 μM did not induce any noticeable morphological abnormality in the embryonic and larval zebrafish, the heart rate was increased as measured by ImageJ TSA method. Moreover, adrenal/interrenal steroidogenesis in larval zebrafish, analyzed by whole-mount 3β-Hsd enzymatic activity and cortisol ELISA assays, was significantly enhanced. During embryonic fin amputation and regeneration, cilostazol treatments led to a subtle yet significant effect on reducing the aggregation of Mpx-expressing neutrophil at the lesion site, but did not affect the immediate injury-induced recruitment and retention of Mpeg1-expressing macrophages.

**Funding:** This work was supported by National Science and Technology Council, Taiwan (www. nstc.gov.tw) 106-2313-B-029-002-MY3 and 109-2313-B-029-002-MY2. The funder had no role in study design, data collection and analysis, decision to publish, or preparation of the manuscript.

**Competing interests:** The authors have declared that no competing interests exist.

## Conclusions

Our results indicate that cilostazol has a significant effect on the heart rate and the growth as well as endocrine function of steroidogenic tissue; with a limited effect on the migration of innate immune cells during tissue damage and repair.

## Introduction

Cilostazol is an anticoagulant clinically used to treat peripheral arterial occlusion and thrombotic complications of coronary angioplasty; and for secondary prevention of stroke [1–4]. It is known to increase the intracellular concentration of cyclic adenosine monophosphate (cAMP), a second messenger in the signaling mediated by G protein coupled receptor, by blocking its metabolism through inhibiting the action of phosphodiesterase (PDE) type 3. Both preclinical and clinical studies support the antiplatelet, antithrombosis and vasodilation actions of cilostazol [5,6]. Owing to these properties, cilostazol has been repurposed as a potential drug for Raynaud's phenomenon [7]. Moreover, it has been proposed as a potential candidate for treating β-hemoglobinopathies, Alzheimer's disease and Covid-19 [8–11]. In the dog and human studies, cilostazol treatments lead to typical cardiovascular effects of PDE3 inhibitors such as enhancement in the heart rate, myocardial contractile force and ventricular automaticity [12,13]. While increasing the concentration of cAMP is known to modulate a multitude of responses in the endocrine and immune systems, how the cilostazol treatment regulates these two systems remain less clear. Understanding how cilostazol impacts the endocrine and immune systems will help us to better address the functions and side effects of cilostazol in the re-purposing studies.

Cilostazol has been found to inhibit secretion of the stress response hormone catecholamine (epinephrine + norepinephrine) in the bovine adrenal chromaffin cells through inhibiting the Ca2+ movement [14]. However, whether and how cilostazol regulates the function of adrenal cortex remains unclear. PDE2, PDE8 and PDE11 are thought be the major PDEs involved in the cAMP-dependent adrenal steroidogenesis in the mammalian model [15–17]. Nevertheless, a few studies pointed out a possible role of PDE3 and its regulator for modulating the adrenal steroidogenesis. In human adrenocortical NCI-H295 cells, multiple PDE isoforms including PDE3A and PDE3B are down-regulated by p54(nrb)/NONO expression which is essential for adrenocorticotropin response, cAMP production and cortisol synthesis [18]. Moreover, leptin downregulates adrenocorticotropin/cAMP signaling, in NCI-H295 cells, possibly through phosphatidylinositol 3-kinase/Akt and PDE3 [19]. On the other hand, corticosterone synthesis in a neonatal hypoxic-ischemic brain injury rat model is inhibited by granulocyte-colony stimulating factor through JAK2 and PDE3B dependent pathway [20]. As the level of corticosteroid is closely associated with the immune regulation [21], it is of interest to examine whether the PDE3 inhibitor cilostazol affects the function of adrenal cortex.

In the previous *in vitro* preclinical studies, cilostazol has been tested to exert anti-inflammatory effects on effector cells of innate immunity. In the LPS-activated murine RAW264.7 macrophages, cilostazol attenuates cytokine expression, and inhibits high mobility group box 1, NF-kB and PAI-1 through activating AMPK/HO-1 [22,23]. Cilostazol suppresses LPS-induced PU.1-linked TR4 expression and TLR2-mediated IL-23 production in synovial macrophages isolated from patients with rheumatoid arthritis [24,25]. In the human monocyte-derived dendritic cells, cilostazol inhibits the production of IL-23 potentially by an AMPK-dependent pathway [26]. However, how cilostazol regulates the *in vivo* behavior of innate immune cells during the process of tissue injury and repair remain unclear.

The tissue transparency, speedy development and a high conserved genome with those in mammals make the zebrafish a robust model for biomedical research. The zebrafish interrenal gland is a functional homolog of mammalian interrenal gland, with highly conserved developmental program and molecular regulation of organ formation [27,28]. As such, the zebrafish has been established as a teleostean model for hypothalamo-pituitary-adrenal axis and steroidogenesis [29–31]. It is also utilized to examine how glucocorticoids including cortisol, dexamethasone, prednosolone and beclomethasone affect development, immune function and injury-induced regeneration [32–35]. Caudal fin amputation of larval zebrafish has become an *in vivo* model of inflammation and wound repair, due to the unique capability for monitoring the repair process as well as immune cell behavior over a long duration [36,37]. The sterile wounds are similar to well-controlled surgical wounds which lack complications of infections. It is therefore particularly useful for studying the effects of glucocorticoids on the immune cell recruitment upon tissue injury and repair. Macrophages and neutrophils are both motile phagocytic cells required for the regulation of tissue damage and repair. Neutrophils are known to play a dominant role during the early stages of inflammation, and macrophages soon follow to function in the repair of tissue damage [38]. Macrophages reside in the tissues while neutrophils typically circulate in the blood, and both can be readily detected and quantified by histochemical or reporter methods in the zebrafish embryo [39]. Zebrafish macrophages and neutrophils start to function during embryonic development, capable of phagocytosis as early as 28 to 30 hours post fertilization (hpf) [40–42].

In this study, we first verified a dose-dependent effect of cilostazol on accelerating the heart rate, further supporting that cilostazol exerts similar pharmaceutical actions in the zebrafish as in mammals. Using the embryonic zebrafish, we then examined whether and how cilostazol affects the development and function of steroidogenic interrenal tissue. Next, we investigated the effect of cilostazol treatments on neutrophils and macrophages during the process of amputation-induced fin regeneration. To our knowledge, this study is the first to demonstrate that cilostazol affects the steroidogenic tissue and endogenous glucocorticoid synthesis, and may therefore impact the glucocorticoid-regulated physiological processes.

## Materials and methods

### Zebrafish husbandry and egg collection

Zebrafish (*Danio rerio*) were raised according to standard protocols [43]. Embryos were obtained by natural crosses of wild-type and transgenic fish, cultured at 28˚C, and staged as described in [44]. The *golden* [45], *Tg(mpx:EGFP)* and *Tg(mpeg1:mCherry)* strains were obtained from Taiwan Zebrafish Core Facility, and the *citrine* strain [46] provided by Prof. Yung-Jen Chuang (National Tsing-Hua University, Taiwan). All experimental procedures on zebrafish were approved by the Institutional Animal Care and Use Committee of Tunghai University (IRB Approval NO. 105–28) and carried out in accordance with the approved guidelines. Anaesthesia of embryos were performed by treating with 0.025% aminobenzoic acid ethyl ester (tricane, Sigma). The embryos were euthenized by rapid chilling prior to histological assays, or at the end of the experiment.

### Chemical treatments

Stock solutions of 20 mg/mL cilostazol (Otsuka Pharmaceutical) were prepared in dimethyl sulfoxide (DMSO, Sigma). Working solutions were freshly prepared by diluting the stock solutions with aerated egg water. The vehicle control contained DMSO at 0.18%. Embryos were placed into these working solutions and cultured at a density of about 30 individuals per 10 ml in a glass beaker. For heart beat measurements, cortisol enzyme-linked immunosorbent assay

(ELISA) and whole-mount 3-β-Hydroxysteroid dehydrogenase /Δ5–4 isomerase (3β-Hsd) enzymatic activity assays, the cilostazol treatments commenced immediately after the completion of gastrulation (10 hpf). For the tail fin amputation experiments, the cilostazol treatments were started at 1 day post-fertilization (dpf).

## Heart rate measurement

The assessment of heart rate was performed by the ImageJ TSA (Temporal Series Analysis) method in [47,48] with modifications. Live images of the zebrafish embryo aged at 31 hpf, 2 dpf and 3 dpf were acquired by a SZX12 microscope (Olympus) equipped with a digital camera and SZX2-ILLTS for contrast-enhancing oblique illumination. Videos were recorded in MP4 format at a frame rate of 25 frames per second by using the ToupLite software. The MP4 videos were converted by VirtualDub2 to the AVI format, and subsequently analyzed by Fiji ImageJ. Region of interest (ROI) was selected based on image contrast with high dynamic pixel changes within the heart region. A time series analyzer V3 plugin (Available online: https:// imagej.nih.gov/ij/plugins/time-series.html) was used to analyze the dynamic pixel changes in the selected ROI. The table data were exported and saved in Excel, and frame was converted into time by dividing the frame rate. Heartbeat time interval was calculated by subtracting two consecutive time points. Beats per minute was obtained by dividing one minute (60 seconds) by the time interval.

## Whole-mount 3β-Hsd enzymatic activity assay and steroidogenic cell counting

Chromogenic histochemical staining of 3-β-Hsd enzymatic activity was performed on whole embryos essentially according to the described protocol in [49], except that no phenylthiourea treatment for depigmenting was included prior to the assay. Stained embryos from the 3$\beta$-Hsd activity assay were cleared in 50% glycerol in phosphate-buffered saline (PBS) and subjected to yolk sac removal. The specimens were photographed as ventral flat-mount under Nomarski optics on an Olympus BX51 microscope system. Interrenal cell counting on the ventral flat mount images of zebrafish at 3 days post fertilization (dpf) after staining for 3-β-Hsd enzymatic activity was based on what is described in [30]. The multi-point tool of Fuji ImageJ was used for counting the stained steroidogenic cells.

## Cortisol extraction and ELISA

The treated embryos were harvested at 5 dpf for cortisol ELISA assays. The cortisol extraction was performed following the protocol described in [50] with modifications. A group of 30 treated embryos pooled as one sample for ELISA was immobilized by ice-cold water and frozen in liquid nitrogen, and 3 ELISA samples were collected from each respective treatment. The samples were homogenized using a pellet mixer (Dr. Owl) and added with ethyl acetate for the collection of supernatant. The supernatant was evaporated using nitrogen gas, and lipid-containing extracted samples were dissolved in 60 μL of 0.2% BSA in PBS.

For the determination of whole-embryo cortisol levels, a cortisol ELISA kit (Salivary Cortisol Enzyme Immunoassay Kit, Salimetrics) was used according to the manufacturer's instructions. Cortisol concentration values were obtained by performing 4 parameter logistic regression of the absorbance readings against a standard curve on a plate reader (TECAN Sunrise).

## Tail fin amputation

Tail fin amputation was performed based on the method described in [51] with modifications. Treated embryos at 3 dpf were anesthetized and amputated at the posterior edge of the notochord with a 1 mm sapphire blade (World Precision Instruments) on 2% agarose-coated dishes under a Nikon SMZ1500 stereomicroscope. After surgery, the embryos were immediately transferred to freshly prepared solutions for continuous chemical or control treatments.

## Detection and imaging of neutrophils by transgenic reporter and enzymatic activity assays of myoloperoxidase (mpx)

The expression and enzymatic activity of neutrophils were detected by using the *Tg(mpx:EGFP)* line as described in [52] and Peroxidase Leukocyte kit (Sigma) respectively. The *Tg(mpx:EGFP)* embryos subject to chemical treatments and fin amputation were anesthetized immediately before imaging. Fluorescent live imaging of *Tg(mpx:EGFP)* embryos was performed by the Axio Observer Z1 microscope (Zeiss) equipped with the Axiovision SE64 software. The number of EGFP positive cells in the counting region was analyzed by the Fuji ImageJ.

Staining of mpx activity by the Peroxidase Leukocyte kit was performed according to the method described in [53] with modifications. Chemical treated and amputated embryos were subject to rapid chilling and fixed by 2% paraformaldehyde in PBS (PFAT) for overnight. After washing by PBS containing 0.1% Tween 20 (PBST) and then trizmal buffer containing 0.1% Tween 20 (TT), the staining reaction was carried out in TT buffer containing 0.015% hydrogen peroxide and 1.5 mg/mL peroxidase indicator reagent at 37˚C. The reaction was stopped by washing with PBST and the stained embryos wre post-fixed in 2% PFAT. The fixed embryos were washed with PBST and cleared in 50% glycerol/PBS before being photographed by an Olympus SZX12 microscope.

## Detection and imaging of macrophages

To detect the macrophage cells recruiting to the amputation site, embryos of *Tg(mpeg1: mCherry)* were subject to the chemical treatments and amputations, and fixed with 1% PFAT at the indicated time points for temporal analysis. The fixed embryos were washed with PBST and cleared in 50% glycerol/PBS, before being photographed with a LSM510 confocal microscope equipped with LSM 3.5 software (Zeiss). The mCherry fluorescent image marking macrophages was recorded by using 3D z stack and projections.

## Quantification of neutrophils, macrophages and the caudal fin area following amputation

The immune cells near the injury site were quantified by measuring the number of those neutrophils or macrophages present in the counting region as indicated in Figs 4E and 6E. The tail fin area of fixed and mounted larval samples was measured from the caudal end of the notochord to the edge of the regenerating fin by using Image J software, and expressed as arbitary units.

## Statistical analysis

Statistics were performed with the R software (Version 4.2.3), and graphs were made with GraphPad Prism 8. All quantitative data are presented as mean ± standard error of the mean (SEM). For comparing multiple groups of data, the normality and homogeneity of variances were analyzed by the Shapiro-Wilk test and the Levene's test respectively. For normally distributed data with homogeneity of variance, ANOVA was used to compare the means of the

groups; and the Duncan's, Scheffé or Tukey's tests for post-hoc analysis. For non-normally distributed data with homogeneity of variance, the Kruskal-Wallis test was used to compare the means of the groups; and the Dunn test for post-hoc analysis. For data that lack homogeneity of variance, the Welch's ANOVA was used to compare the means of the groups; and the Games-Howell test for post-hoc analysis.

## Results

### The cilostazol treatment enhances heart rate in the zebrafish in a dose-dependent manner

In order to test the pharmaceutical actions of cilostazol on the general morphology and development of zebrafish, the embryonic zebrafish were treated with various concentrations of cilostazol from 10 hpf onwards, when the process of organogenesis initiates. Dosages of cilostazol treatments at 10, 30, 50 and 100 μM did not cause noticeable morphological effect on the developing zebrafish, implicating a low toxicity of cilostazol on the growing tissues. Moreover, the heart rate of developing zebrafish was not affected by various concentrations of cilostazol at 31 hours post fertilization (hpf) (Fig 1). At 2 dpf, a 1.1 fold increase in heart rate was shown

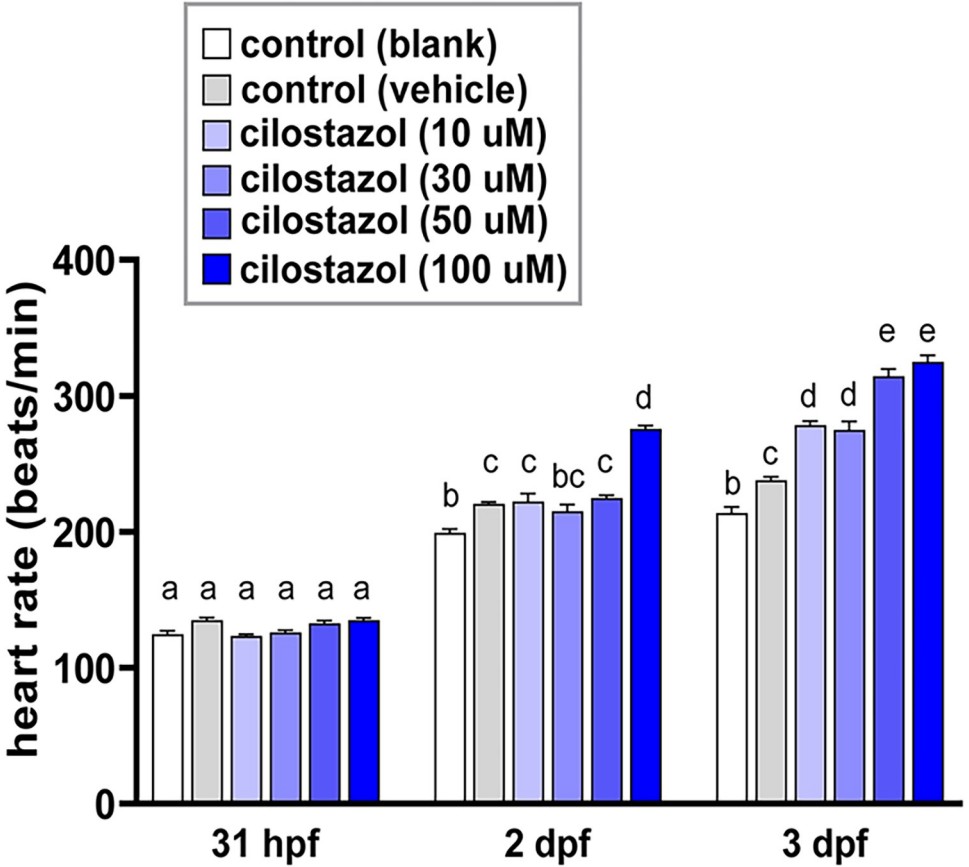

**Fig 1. The effect of cilostazol on the developing zebrafish at various stages.** The wild-type embryos of zebrafish were treated with increasing concentrations of cilostazol as well as the blank or vehicle controls from 10 hpf onwards. For each treatment group, 10 embryos were subject to the oblique illumination microscopy and measurement of heart rate by ImageJ TSA method at 31 hpf, 2 dpf and 3 dpf respectively. Histograms with different letters above them are significantly different (ANOVA and Tukey's multiple comparisons test, $P < 0.05$). The representative video clips for the heartbeat of 3 dpf embryos in the blank control, vehicle control or 100 μM cilostazol-treated groups are shown in S1, S2 and S3 Videos respectively.

in the vehicle control group as compared to the blank control group, and there was no significant difference in heart rates among the groups of vehicle control and cilostazol treatments at 10, 30 and 50 μM. Nevertheless, the cilostazol treatment at 100 μM resulted in an evident increase in heart rate as compared to other treatment groups. Although the DMSO vehicle control treatment also caused a 1.1 fold increase in heart rate as compared to the blank control at 3 dpf, a stage when the zebrafish starts a transition from embryos to larvae; a significant increase in the heart rate was detected in all cilostazol treatment groups as compare to either blank or vehicle control. Lower cilostazol dosages at 10 and 30 μM led to enhanced heart rates at 3 dpf with no significant difference between two treatments. Further elevated heart rates at more than 300 beats per minute were detected in those groups treated with high cilostazol dosages at 50 and 100 μM, with no significant difference between two groups. In summary, cilostazol leads to an increased heart rate starting from 2 dpf by the highest dosage at 100 μM, and all dosages of cilostazol enhance heart rates at 3 hpf; with higher dosages at 50 and 100μM cause a more evident effect. This result is consistent with the tachycardia phenomenon seen in cilostazol-treated human adults and the dose-dependent increase of heart rate in healthy dogs treated with cilostazol [54,55], implicating that the pharmacological effects of cilostazol on developing zebrafish may resemble its effects on adult mammals.

## The cilostazol treatment leads to an increase of steroidogenic cells as well as cortisol levels in zebrafish

It is known that the activation of protein kinase A (PKA) by cAMP increases the gene expression of StAR (steroidogenic acute regulatory protein), which is crucial for steroidogenesis in the adrenal cells. However, whether and how cilostazol, which enhances cAMP production by limiting the cAMP-degrading activity of PDE3 [56], affects the cellular physiology of the adrenal gland *in vivo* remains unclear. We therefore used the zebrafish embryo to test whether cilostazol affects development and function of the cortisol-producing steroidogenic tissue. Histochemical staining of 3β-Hsd activity has been routinely used to detect differentiated steroidogenic cells in isolated tissues from mammals as well as teleosts [57,58], and can be applied to whole-mount embryos and larva of zebrafish [49]. Embryos subject to immersions in cilostazol- or vehicle-containing egg water were assayed at 3 dpf, a stage when the organogenesis of interrenal gland is completed and ready for stress-initiated responses [29,30]. As pigment formation at the dorsal side of zebrafish pronephros is evident at such stage [59] and often obscures the results of 3β-Hsd staining, we used two low pigment-producing strains, *golden* [45] and *citrine* [60]; and the embryos stained with whole-mount 3β-Hsd activity assay were deyolked and visualized from the ventral side. For both *golden* and *citrine* strains, the embryos treated with cilostazol at 50 and 100 μM respectively demonstrated a significant dosage-dependent increase of steroidogenic interrenal cells, as compared to those treated with vehicle control (Fig 2A–2C).

Cortisol is the principal steroidogenic hormone for mediating stress response in teleosts [61]. To further estimate whether the increase of differentiated steroidogenic interrenal cells contributes to higher amount of steroidogenic hormone secretion, whole-body cortisol extraction and cortisol ELISA assays were applied to the larval zebrafish groups treated with different concentrations of cilostazol or control solutions. As compared to either blank or solvent controls, the cortisol amount in developing zebrafish treated with cilostazol at 10 or 30 μM did not cause increased cortisol levels; yet those treated with cilostazol at 50 or 100 μM led to a significant increase of cortisol amounts at 5 dpf. It is consistent with the increased steroidogenic interrenal cells at 3 dpf old embryos treated with cilostazol at 50 or 100 μM, as compared to those with the solvent control (Fig 2). While the interrenal steroidogenic cells was increased in

**Fig 2. The effect of cilostazol on the morphology and cell number of steroidogenic interrenal tissue in the zebrafish embryo at 3 dpf.** The embryos of *golden* and *citrine* strains were treated with cilostazol-supplemented egg water at 50 and 100 μM respectively, or with the vehicle control. The treated embryos were harvested at 3 dpf for whole-mount 3βHsd activity staining. (A) The stained interrenal cells (bluish purple signals at the cytoplasm) were detected on the ventral surface of deyolked embryos with anterior oriented to the left, by using Nomarski microscopy. Scale bar, 50 μm. The number of cells positive for 3βHsd activity staining in the treated *golden* and *citrine* embryos is quantified in (B) and (C) respectively. Sample images for the ImageJ quantification of interrenal cells are shown in S1 Fig. Histograms with different letters above them are significantly different (ANOVA and Duncan's multiple test, $P < 0.05$).

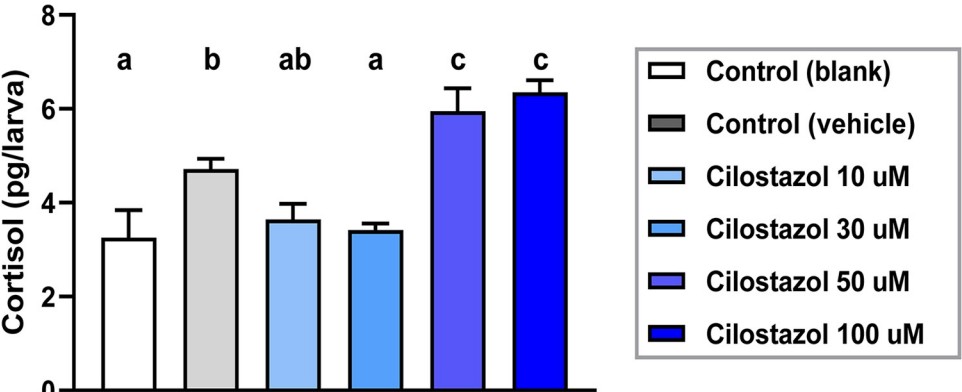

**Fig 3. Whole-body cortisol levels after exposure to varying doses of cilostazol as well as controls from 10 hpf to 5 dpf, and measured by ELISA of larval extracts.** Each bar represents the average of 3 independent treated groups, with each group containing 30 larvae and subject to duplicate ELISA tests. Histograms with different letters above them are significantly different (ANOVA and Duncan's multiple test, P < 0.05).

a dose-dependent manner at 3 dpf, the cortisol amount at 5 dpf however did not vary between 50 and 100 μM of cilostazol treatments. It is noted that the vehicle control containing DMSO at 0.18% caused a 1.4 fold elevation of cortisol production in the 5 dpf embryo as compared to the blank control (Fig 3). It remains unclear why the cilostazol at 30 μM caused a lower cortisol production as compared to the vehicle control group. Nevertheless, it is evident that cilostazol at 50 and 100 μM led to approximately 1.3- to 1.4- fold elevation of cortisol production as compared to the vehicle control. In summary, the analysis of interrenal steroidogenic cell counts (Fig 2) and cortisol production (Fig 3) both support an enhancing effect of cilostazol on steroidogenesis at the concentrations of 50 and 100 μM.

### The effect of cilostazol treatments on the injury-induced recruitment of neutrophils during zebrafish fin amputation and wound healing

It has been shown in the fin amputation-induced model of inflammation that several synthetic glucocorticoid hormones attenuate the migration of neutrophils toward the wounding site, with that of macrophages unaffected [34,62,63]. Whether an elevation of endogenous cortisol levels due to stress or pharmaceutical effects also influence the immune cell accumulation in this wounding-induced inflammation model remains unclear. As our results displayed that cilostazol at 50 and 100 μM enhanced the interrenal steroidogenesis in the zebrafish embryo (Figs 2 and 3), it is of interest to examine the effect of cilostazol on the behavior of innate immune cells during tail fin amputation and regeneration. The tail amputation was performed on zebrafish embryos at 3 dpf, a stage when neutrophils and macrophages are the two types of leukocytes that constitute the innate immune system. The zebrafish larvae after tail amputation were immediately subject to cilostazol or control treatments. For the analysis of neutrophil behavior during the injury-induced inflammation, we used the *Tg(mpx:EGFP)* line which recapitulates the expression of *mpx* gene in the myeloid lineage [52], in a time-course live imaging study (Fig 4A–4D). Prior to amputation, the neutrophil count was very low in the tail fin area of either control or cilostazol treated embryos (Fig 4A and 4C). In the selected tail fin region for counting (Fig 4B), the neutrophil number in all treatment groups were evidently increased at 6 hours post amputation (hpa), with no significant difference among the control groups and the 10 and 30 μM cilostazol treated groups (Fig 4A and 4C). It is noted that the neutrophil number in the 50 μM cilostazol treated group was significantly lower than those in all

treatment groups except that in the 100 μM cilostazol treated group. The neutrophil number in the 100 μM cilostazol treated group was lower than those in the blank control and 30 μM cilostazol treated groups; but not significantly different from those in the vehicle control and 10 as well as 50 μM cilostazol treated groups. At 12 and 24 hpa, the neutrophil number near the tail injury showed no difference among all treatment groups. These results suggest that cilostazol at 50 μM may downregulate the recruitment of *mpx*-expressing cells at 6 hpa, but does not affect the retention of neutrophils at subsequent stages.

Consistent with previous studies in [52], the number of neutrophils displayed a declining trend from 6 to 24 hpa in the blank control group, as well as in all the cilostazol treatment groups (Fig 4D). On the other hand, no statistically significant decline of neutrophil numbers from 6 to 24 hpa was detected in the vehicle control group, implicating a stimulating effect of DMSO on the retention of neutrophils, which however did not cause any significant difference of neutrophil numbers among all treatment groups at 24 hpa (Fig 4C).

Apart from the time course live imaging study by using the *Tg(mpx:EGFP)* line, a whole mount Mpx enzymatic activity-based assay was performed to evaluate the effect of cilostazol on the Mpx activity-positive neutrophils, in the fixed embryos harvested at different time points after amputation (S2 Fig). The Mpx-activity of neutrophils promoted the formation of brown-black insoluable reaction products that can be readily differentiated from the melanin pigments under Nomarski microscopy (S2A Fig). Similar to what was shown in the assay on *Tg(mpx:EGFP)*, the neutrophil counts in all treatment groups were evidently increased at 6 hpa, followed by a declining trend at 12 hpa (S2D Fig). However, a resurgence of Mpx-activity positive cells was noted in both blank and vehicle control groups at 24 hpa. There was no significant difference of neutrophil counts among all treatment groups except for 24 hpa, when the neutrophil counts detected in the blank control (13.2±0.9) and vehicle control (13.2±1.8) groups were significantly higher than those in 50 μM (8.0±1.1) and 100 μM (7.7±1.2) cilostazol treatment groups (S2B Fig). To check whether cilostazol specifically affects the retention of neutrophils in the wounded area rather than their total number, Mpx activity-positive cells in the entire tail region at 24 hpa were counted (S3 Fig). No significant difference was found among all the treatment groups, indicating a specific effect of cilostazol on the neutrophil in the wounded area at 24 hpa.

It remains unclear why the trend of neutrophil dynamics and the effect of cilostazol treatments appeared divergent between *mpx:EGFP* live imaging and Mpx enzymatic activity staining assays. The treated groups of embryos for Mpx activity assays were separately harvested at each stage without repeated anesthesia required for the time course study on *Tg(mpx:EGFP)*, which may cause different physiological backgrounds especially at late stages. Moreover, whether *mpx*-expressing neutrophils marked by *Tg(mpx:EGFP)* are fully identical to functional maturated Mpx enzymatic activity positive population at all stages is to be considered. It is noted that at 6 hpa, the cell counts of *mpx:EGFP* positive neutrophils (Fig 4B) appeared to be generally higher than those of the Mpx activity-stained neutrophils (S2B Fig). Nevertheless, our results in Fig 4 support that cilostazol at 50 μM influences the recruitment but not subsequent retention of *mpx*-expressing neutrophils near the injury site, while those in S2 Fig implicate a possible suppressing effect of cilostazol at 50 and 100 μM on the number of mature Mpx activity-positive cells in the regenerating fin. Whether physiological or genetic backgrounds lead to this discrepancy may await future investigation.

## The effect of cilostazol treatments on the wound healing of zebrafish caudal fin-folds

Apart from evaluating the effect of cilostazol treatments on the neutrophil cell behavior following tail fin injury, we have also checked whether the growth of regenerating wounds was

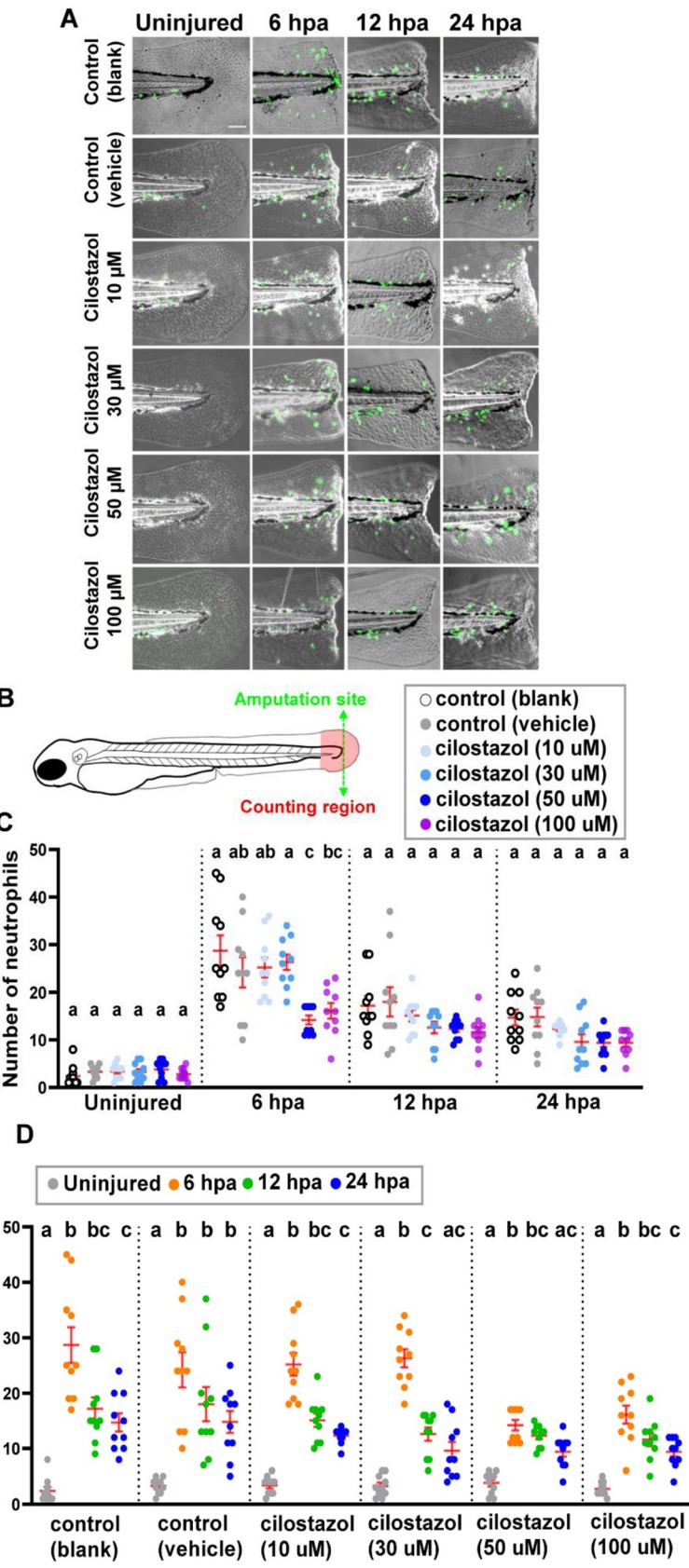

**Fig 4. The effect of cilostazol treatment on the accumulation of Mpx-expressing neutrophils during fin amputation and regeneration.** (A) The embryos of *Tg(mpx:EGFP)* line were subject to treatments of cilostazol at 10, 30 50 and 100 μM respectively, or blank and vehicle controls, from 24 hpf onwards. 10 embryos were contained in each treatment group for the time course analysis. The amputation of caudal fin fold was performed on the zebrafish embryos at 3 dpf, and live imaging was used to visualize and record the GFP-expressing neutrophils in the tail region of embryos prior to amputation as well 6, 12 and 24 hpa respectively. (B) The green dotted line in the schematic diagram indicates the site of resection, and the area in-between levels of the anterior edge of pigment gap and the posterior edge of regenerating fin was selected for neutrophil quantification. (C) A comparison of neutrophil accumulation in the selected counting region, among different treatment groups at the uninjured fins; or at regenerating fins at 6, 12 and 24 hpa respectively. For each time point, differences in the number of neutrophils were compared among various treatment groups. Kruskal-Wallis followed by Dunn's test was performed for the analysis of uninjured and 6 hpa groups. Welch's ANOVA followed by Games-Howell test was performed for the analysis of 12 and 24 hpa groups. Columns of data with different letters above them are significantly different (P < 0.05). (D) A comparison of neutrophil accumulation in the wounded fin area among different time points after fin amputation, in each treatment group. Kruskal-Wallis followed by Dunn's test was performed for the analysis of blank and vehicle control groups, as well as for the 10, 30 and 50 μM cilostazol treated groups. Welch's ANOVA followed by Games-Howell test was performed for the 100 μM cilostazol treated group. Columns of data with different letters above them are significantly different (P < 0.05).

affected by cilostazol. The wound healing process of amputated zebrafish larval fin-fold involves the formation of wound epithelium starting at 6 hpa, and the blastema formation as well as active cell proliferation evident from 24 hpa onwards [64,65]. Consistent with earlier studies, epithelial cells surrounding the amputation stump contracted and sealed the wound at 6 hpa in the blank control group, and this process was not affected in either blank control or cilostazol treatment groups (Figs 4A and S2A). As the tricane anesthesia used for live imaging assays may affect embryonic physiology such as heart rate [66], samples in the fin amputation experiment in S2 Fig which were not exposed to tricane treatments were used for evaluating whether embryonic fin regeneration is affected by cilostazol treatments. As the growth of regenerating caudal fin-fold was measured by quantifying the tail area posterior to the caudal end of the notochord, there was an evident increase from 12 to 24 hpa in all treatment groups (Fig 5B). Meanwhile, there was no significant difference of quantified caudal fin-fold area among all treatment groups at 6, 12 and 24 hpa (Fig 5A). It is therefore concluded that the wound healing in the zebrafish larval fin model is generally not affected by the cilostazol treatment.

## The effect of cilostazol treatments on the injury-induced recruitment of macrophages during zebrafish fin amputation and wound healing

In order to evaluate the influence of cilostazol treatments on the behavior of macrophages in the inflammatory model of zebrafish fin amputation, the *Tg(mpeg1:mCherry)* line was used to detect the presence of macrophages. In zebrafish, the *mpeg1* gene is expressed as early as 28 hpf by both M1 and M2 types of macrophages [37,67]. The *Tg(mpeg1:mCherry)* embryos were subject to cilostazol or control treatments from 1 dpf onwards, and the treatments were continued immediately after the tail amputation at 3 dpf and lasted till the time of sample collections. Macrophages in the caudal fin area at different time points after amputation were examined by confocal microscopy (Fig 6A). In all treatment groups, the macrophage count was low in the uninjured tail fin and evidently increased at 6 hpa, when the macrophages were recruited to and aggregated near the wounding site. While a comparison was made among all treatment groups for each time point, no significant difference of macrophage counts among treatment groups was revealed at 12 and 48 hpf (Fig 6B and 6C). The 50 μM cilostazol treatment at 6 hpa caused a significantly higher macrophage count (37±3.1) as compared to the blank control (21.9±2.9) but not to the vehicle control group (35.1±3.8); and the 100 μM cilostazol treatment led to a macrophage count (25.3±4.3) not significantly different from either

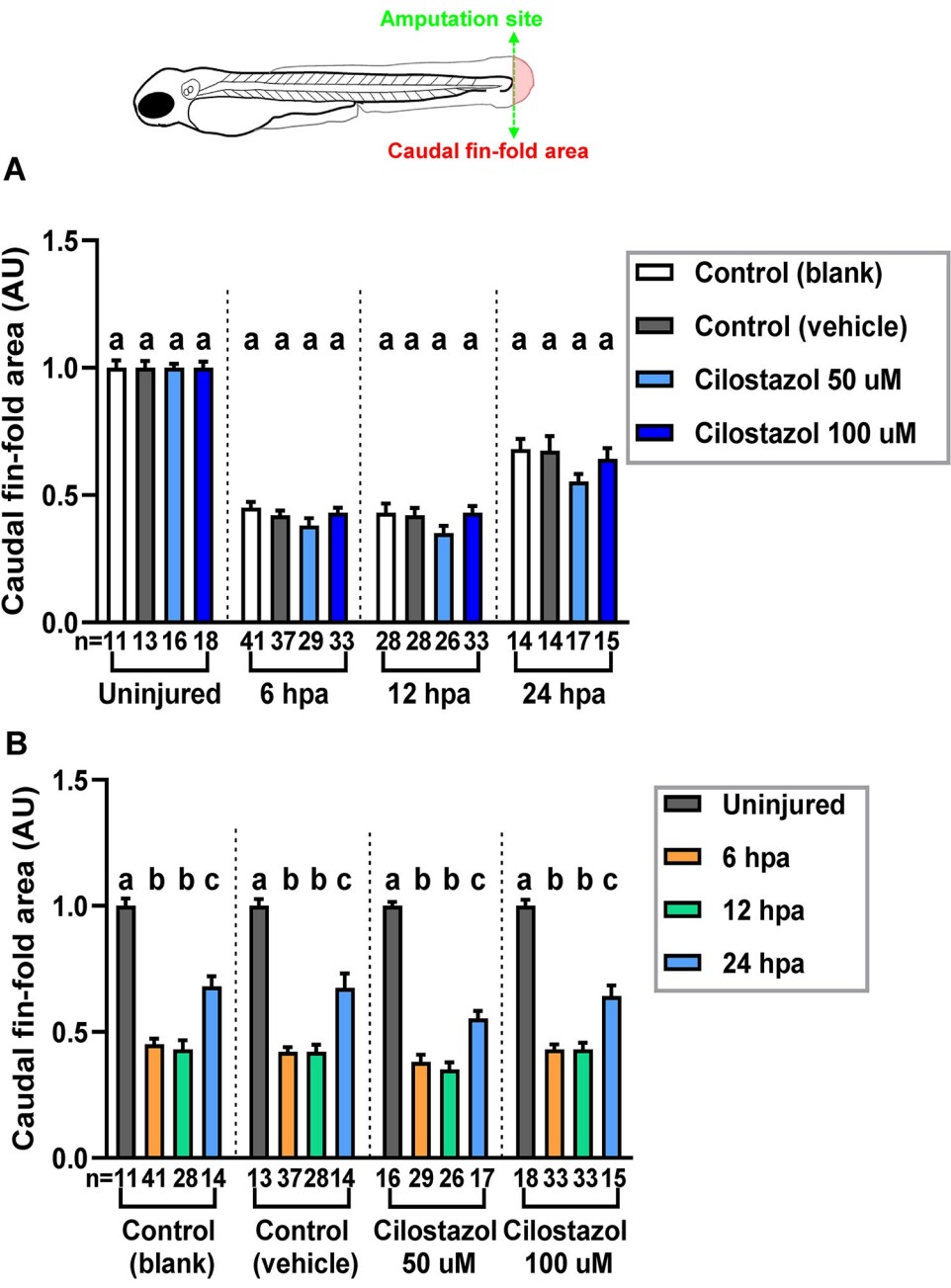

**Fig 5. The effect of cilostazol on the regeneration of caudal fin fold zebrafish.** The embryos subject to the cilostazol treatments and fin amputation as in S2 Fig. were analyzed to evaluate whether cilostazol affects the regeneration of fin fold post-injury. The regenerated fin is defined as the surface area of the fin fold posterior to the amputation site, as viewed laterally. (A) The regenerated caudal fin area of the injured embryos under different treatments are quantified and compared at 6, 12 and 24 hpa respectively, following the amputation at 3 dpf. The uninjured fin areas posterior to the hypothetical site of resection are also quantified and compared among different treatments at 3 dpf. ANOVA analysis followed by Scheffé test was performed for comparisons among different treatment groups of uninjured fins, and injured fins at 24 hpa. Kruskal-Wallis followed by Dunn's test was performed for comparisons among different treatment groups of injured fins at 6 and 12 hpa. Histograms with different letters above them are significantly different ($P < 0.05$). (B) The regenerated fin area for each type of cilostazol or control treatments is compared among 6, 12 and 24 hpa following amputation at 3 dpf; and contrasted with the caudal fin area of uninjured embryos at 3 dpf. Kruskal-Wallis followed by Dunn's test was performed for the analysis of blank control group. Welch's ANOVA followed by Games-howell test was performed for the analysis of vehicle control and 50 μM cilostazol treatment groups. ANOVA analysis followed by Scheffé test was performed for the analysis of 100 μM cilostazol treatment group. Histograms with different letters above them are significantly different ($P < 0.05$).

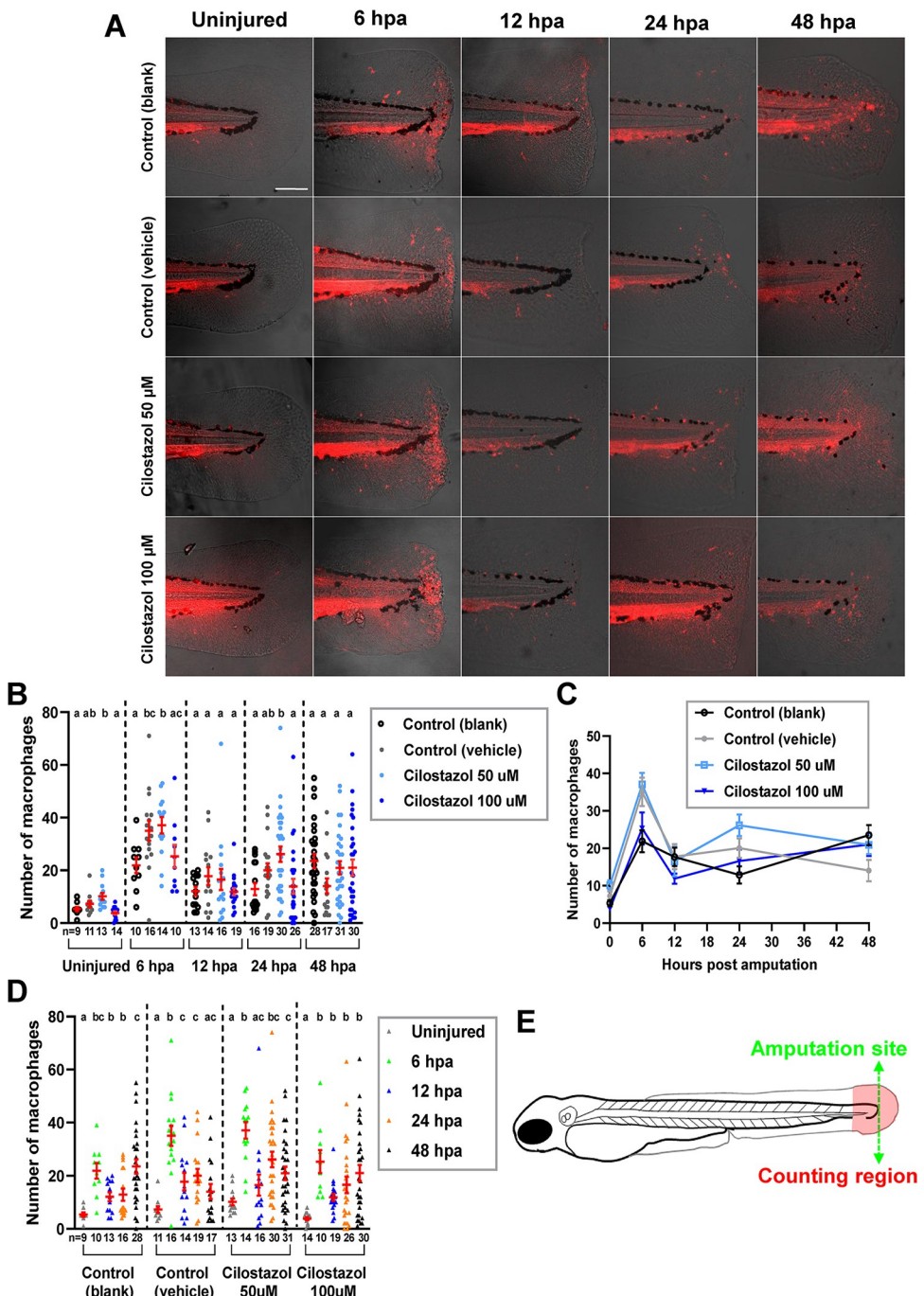

**Fig 6. The effect of cilostazol treatment on the accumulation of Mpeg1-expressing macrophages during fin amputation and regeneration.** (A) The embryos of *Tg(mpeg1:mcherry)* line were subject to treatments of cilostazol at 50 and 100 μM respectively, or blank and vehicle controls, from 24 hpf onwards to the time of harvest. The amputation of caudal fin fold was performed on the embryos at 3 dpf, which were subsequently fixed at 6, 12, 24 and 48 hpa respectively for macrophage quantification. The uninjured fins at 3 dpf in different treatment groups were also collected for comparisons. The amputation site (green dotted line) and the macrophage counting region (in-between levels of the anterior edge of pigment gap and the posterior edge of regenerating fin) are depicted in the schematic diagram in (E). (B) A comparison of macrophage accumulation in the selected area of counting, among different treatment groups at the uninjured fins at 3 dpf; or at regenerating fins at 6, 12 and 24 hpa respectively. Kruskal-Wallis followed by Dunn's test was performed for the multiple comparisons among different treatment groups prior to injury, as well as at 12, 24 and 48 hpa. ANOVA analysis followed by Duncan's test was performed for multiple comparisons among different treatment groups at 6 hpa. Columns of data with different letters above them are significantly different

(P < 0.05). (C) A line chart showing the temporally dynamic changes of macrophage accumulation in the wounded fin area, in different treatment groups as shown in (B). (D) A comparison of macrophage accumulation in the wounded fin area among different time points, in each treatment group as shown in (A). Welch's ANOVA followed by Games-howell test was performed for the analysis of blank control as well as 50 and 100 μM cilostazol treatment groups. Kruskal-Wallis followed by Dunn's test was performed for the analysis of vehicle control group. Columns of data with different letters above them are significantly different (P < 0.05).

blank or vehicle control groups. It is noted that at 6 hpa the vehicle control treatment caused a 1.6 fold elevation of the macrophage number as compared to the blank control, while no significant influence on the macrophage number was detected at other stages. It is known that DMSO can either enhance or suppress innate immunity in a highly context-dependent manner, depending on the duration and concentration of treatments [68]. At 24 hpa, the 50 μM cilostazol treatment caused a significantly higher macrophage count (26.1±2.9) as compared to blank control (12.9±2.3) but not to the vehicle control group (20.1±2.6); and the macrophage count in the 100 μM cilostazol treated group (21.0±3.1) is not significantly different from those of both control groups.

While the migration of macrophages toward the wounding site was most evident at 6 hpa in all treatment groups, a retention of macrophages in the caudal fin-fold area was detected at the following stages up to 48 hpa (Fig 6A and 6D). In the blank control group, the macrophage count was not significantly decreased at the stages following 6 hpa; and a similar trend was noted in the 100 μM cilostazol treated group. A slight decrease of macrophage counts at the stages following 6 hpa was detected in the vehicle control and 50 μM cilostazol treated groups. In terms of the migration and retention of macrophages in the fin amputation model, our results therefore do not suggest a clear influence by the cilostazol treatments.

## Discussion

To our knowledge, this is the first study to investigate the impact of cilostazol on growth and endocrine function of the steroidogenic tissue. Meanwhile, our experiments have also helped to elucidate the effects of cilostazol on the injury-induced inflammatory and regenerative processes *in vivo*. The results in this study implicate that cilostazol may exert a similar effect on human fetal and adult adrenal glands. The adrenal gland is known to be an organ undergoing constant regeneration, and the adrenal cortex is continuously renewed by stem cells throughout the adult life [69,70]. As development and renewal of the adrenal cortex are regulated by common regulatory factors such as Hedgehog, Wnt and ACTH/PKA signaling pathways, it is possible that cilostazol administration may affect the progenitor and stem cell populations of the adrenal cortex, thus altering the adrenal function which is critical for homeostasis in the body.

In our study, the heart rate, interrenal tissue growth as well as cortisol secretion in zebrafish were enhanced in the cilostazol treatment experiments. Natural and synthetic glucocorticoids are known to suppress pro-inflammatory macrophages as well as induce anti-inflammatory monocytes and macrophages, thus impeding the expansion of inflammation [21]. Nevertheless, the cortisol regulation of innate immunity in humans can be both pro-inflammatory and anti-inflammatory [71]. Interestingly, the LPS-induced inflammatory response in RAW264.7 cells, associated with activated NF-κB and MAPK pathways, can be attenuated by either cortisol [72] or cilostazol [22], implying that an increase of cortisol due to cilostazol treatments may be involved in the suppression of pathogen-induced inflammation.

*In vitro* experiments in earlier studies indicate that cilostazol suppresses the pro-inflammatory cytokines in the LPS-activated murine RAW264.7 as well as human synovial macrophages. [22–24]. However in our *in vivo* experiments, cilostazol exerted no negative effect on

either macrophage recruitment or regeneration induced by the fin amputation (Figs 5 and 6). In the fin amputation model, macrophages recruited toward the wound site are essential for the tissue regeneration [39,73]. The unaffected macrophage accumulation in the wound area of cilostazol-treated fish was therefore consistent with the normal regeneration of amputated fin. In the scenario of sterile injury, the inflammatory response is known to be essential for the repair process [74]. The initial phase of the inflammatory response is marked by the presence of pro-inflammatory cytokines and the infiltration of immune cells into the wound area. In the subsequent resolution phase, the macrophages switch toward a pro-regenerative phenotype with a decline of cytokine secretion; which is followed by the final phase of inflammation marked by tissue regeneration. Our results indicate that the migratory behavior of neutrophils but not macrophages is affected by cilostazol during tissue damage, which is consistent with earlier findings in [34] that the glucocorticoid suppresses migration of neutrophils but not macrophages. It may also suggest that the effects of cilostazol on innate immune cells during the trauma/tissue damage are distinct from those under exogenous pathogen infections. In zebrafish, the innate immune response to microbial infections can be mounted as early as 28 hpf, by the phagocytic activity of neutrophils and macrophages which display conserved transcriptional signatures resembling those in mammals [40]; while the adaptive immunity becomes mature at around 4–6 weeks post-fertilization [75]. Therefore, our results represent the effects of cilostazol on innate immune cells in the absence of T or B cells, during tissue damage. Cilostazol has been tested beneficial in terms of suppressing encephalitogenic T-cell responses and enhancing regulatory T cell activity [76]. In contrast, our results suggest a limited influence of cilostazol on injury-induced inflammation and regeneration in the immuno-deficient scenario.

While the increased cortisol production due to cilostazol treatments might contribute to the mild suppressing effect of cilostazol on neutrophil migration, it is not to be ruled out that cAMP levels might also affect the neutrophil behavior in the zebrafish fin amputation model. cAMP is known to regulate the migration of neutrophils in a complex and dynamic way. cAMP promotes neutrophil migration by regulating cytoskeletal elements through PKA activation [77]. However, another study suggests that cAMP inhibits neutrophil migration by disrupting polarity and chemotaxis [78]. Specific for the fin amputation model in zebrafish, it awaits future research to investigate any possible combined impact of cortisol and cAMP on the neutrophil migration.

Although it is generally uncertain whether cilostazol is safe and effective for use in children, a clinical trial testing the effects of cilostazol for the treatment of juvenile Raynaud's Phenomenon, a blood vessel disorder in children, has been completed with no significant beneficial effects discovered [79]. Also, cilostazol has been tested effective in a rat model of prenatal valproic acid-induced autism spectrum disorder [80] and hence proposed as an adjunctive therapy for the children with autism spectrum disorder [81]. However, our study suggests potential adverse endocrine effects of cilostazol for use during developmental stages, which need to be carefully assessed. The maximal level of cortisol increase caused by the cilostazol treatment was 1.4 fold as compared to the solvent control (Fig 3). On the other hand, direct early exposure of zebrafish embryos to the cortisol-containing medium, in the study by Hartig et. all, leads to a maximal 1.3 fold elevation of whole-body cortisol amount; which culminates in upregulated basal expressions of pro-inflammatory genes during the later adult stage accompanied by defective adult tailfin regeneration [32]. In human and animal studies, early life stress is known to cause long lasting effects on developing and function of the brain, even causing persistent mental disorders such as anxiety symptoms [82]. Early life stress in the form of childhood adversity is linked to epigenetic regulation of glucocorticoid receptor in the brain, as well as altered neuronal gene expressions due to sustained DNA hypomethylation

[83,84]. In the zebrafish, long lasting anxiety following early life stress is mediated by cortisol and glucocorticoid signaling [85]. The studies above suggest that the hyperactivated glucocorticoid signaling during early life causes persistent effects on the neuronal as well as immune functions lasting to adulthood. Therefore, the results of our study implicate that exposure to cilostazol during early development may lead to pro-inflammatory adults with changes in mental function, which remains to be investigated in animal models.

## Supporting information

**S1 Fig. Sample images for the ImageJ quantification of 3β-Hsd activity-positive interrenal cells.** The representative images of *golden* (A, A1, A2) or *citrine* (B, B1, B2) embryos treated with 100μM cilostazol respectively as in Fig 2 are shown as examples. Panels (A, B) demonstrate the cell counting by ImageJ. (A1, A2, B1, B2) In the cases where stained interrenal cells on the ventral surface need to be identified by adjusting the focus, multiple images were taken in order to provide sufficient resolution for the whole interrenal tissue clusters.
(TIF)

**S2 Fig. The effect of cilostazol treatment on the accumulation of Mpx activity-positive neutrophils during fin amputation and regeneration.** (A) The embryonic zebrafish were subject to treatments of cilostazol at 50 and 100 μM respectively, or blank and vehicle controls, from 24 hpf onwards to the time of harvest. The amputation of caudal fin fold was performed on the zebrafish embryos at 3 dpf, which were subsequently fixed at 6, 12 and 24 hpa respectively for whole-mount Mpx enzymatic staining. The uninjured fins at 3 dpf were also stained to show the background level of neutrophils prior to their migration to the wounded region. The green dotted line indicates the site of resection, and the area in-between levels of the anterior edge of pigment gap and the posterior edge of regenerating fin, highlighted by red broken lines, was selected for neutrophil quantification. The amputation site and neutrophil counting region are also depicted in the schematic diagram in (E). (B) A comparison of neutrophil accumulation in the selected area of counting as shown in (A), among different treatment groups at the uninjured fins; or at regenerating fins at 6, 12 and 24 hpa respectively. At each time point, differences in the number of neutrophils were compared among various treatment groups. Columns of data with different letters above them are significantly different (Kruskal-Wallis analysis followed by Dunn's test, $P < 0.05$). (C) A line chart showing the temporally dynamic changes of neutrophil accumulation in the wounded fin area, in different treatment groups as shown in (B). (D) A comparison of neutrophil accumulation in the wounded fin area among different time points after fin amputation, in each treatment group as shown in (A). Kruskal-Wallis followed by Dunn's test was performed for the analysis of blank and vehicle control groups. Welch's ANOVA followed by Games-Howell test was performed for the analysis of 50 and 100 μM cilostazol treated groups. Columns of data with different letters above them are significantly different ($P < 0.05$).
(TIF)

**S3 Fig. The effect of cilostazol treatment on the number of Mpx activity-positive neutrophils in the entire tail region at 24 hpa.** (A) The area used to determine the total amounts of neutrophils present in the whole tail region (posterior to the yolk extension; YE) is indicated by a red dashed box. (B) A comparison of neutrophil accumulation in the counting area as shown in (A) among different treatment groups at 24 hpa, where no statistical difference is observed (ANOVA and Tukey's multiple comparisons test).
(TIF)

**S1 Video. The heartbeat of a representative 3 dpf embryo in the blank control group.**
(MP4)

**S2 Video. The heartbeat of a representative 3 dpf embryo in the vehicle control group.**
(MP4)

**S3 Video. The heartbeat of a representative 3 dpf embryo in the 100 μM cilostazol-treated groups.**
(MP4)

## Acknowledgments

The authors would like to thank Otsuka Pharmaceutical for providing pure cilostazol substance; Dr. Shih-Lei Lai and Mr. Hsing-Wei Liu for helping the macrophage reporter study; Ms. Wei-Leng Chen for excellent assistance in fish culture. We would also like to acknowledge the technical services provided by the Taiwan Zebrafish Technology and Resource Center of the National Core Facility Program for Biotechnology, National Science Council, Taiwan.

## Author Contributions

**Conceptualization:** Wei-Chun Chang, Yung-Jen Chuang, Yi-Wen Liu.

**Data curation:** Wei-Chun Chang, Mei-Jen Chen.

**Funding acquisition:** Yi-Wen Liu.

**Investigation:** Wei-Chun Chang, Mei-Jen Chen, Chung-Der Hsiao, Rong-Ze Hu, Yu-Shan Huang, Yu-Fu Chen, Tsai-Hua Yang, Guan-Yi Tsai, Chih-Wei Chou, Yi-Wen Liu.

**Methodology:** Wei-Chun Chang, Chung-Der Hsiao, Rong-Ze Hu, Yi-Wen Liu.

**Project administration:** Yi-Wen Liu.

**Resources:** Ren-Shiang Chen.

**Supervision:** Ren-Shiang Chen, Yung-Jen Chuang, Yi-Wen Liu.

**Validation:** Yi-Wen Liu.

**Writing – original draft:** Wei-Chun Chang, Yi-Wen Liu.

**Writing – review & editing:** Yung-Jen Chuang, Yi-Wen Liu.

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
