## [Decision Letter · Decision Letter 0]

25 Jan 2023

PONE-D-23-00030The anti-platelet drug cilostazol enhances interrenal steroidogenesis and exerts a scant effect on innate immune responses in zebrafishPLOS ONE

Dear Dr. Liu,

Thank you for submitting your manuscript to PLOS ONE. After careful consideration, we feel that it has merit but does not fully meet PLOS ONE’s publication criteria as it currently stands. Therefore, we invite you to submit a revised version of the manuscript that addresses the points raised during the review process. In particular, please focus on providing sufficient justifications of the claims by clarifying your experiments or by performing the suggested experiments. Otherwise, please delete all upsuported claims. 

We look forward to receiving your revised manuscript.

Kind regards,

Yuk Fai Leung, Ph.D.

Academic Editor

PLOS ONE

Journal Requirements:

"This work was supported by National Science and Technology Council (NSTC) 106-2313-B-029-002-MY3 and 109-2313-B-029-002-MY2 (Taiwan)."

"This work was supported by National Science and Technology Council, Taiwan (www.nstc.gov.tw) 106-2313-B-029-002-MY3 and 109-2313-B-029-002-MY2 . 

The funder had no role in study design, data collection and analysis, decision to publish, or preparation of the manuscript."

Reviewers' comments:

Reviewer's Responses to Questions

**Comments to the Author**

1. Is the manuscript technically sound, and do the data support the conclusions?

Reviewer #1: Partly

Reviewer #2: Partly

2. Has the statistical analysis been performed appropriately and rigorously? 

Reviewer #1: No

Reviewer #2: Yes

3. Have the authors made all data underlying the findings in their manuscript fully available?

Reviewer #1: Yes

Reviewer #2: No

4. Is the manuscript presented in an intelligible fashion and written in standard English?

Reviewer #1: Yes

Reviewer #2: Yes

5. Review Comments to the Author

Reviewer #1: In this study the authors attempt to elucidate the cilostazolrenalsteroidneutrophil recruitment axis, where cilostazol can lead to decreased neutrophil recruitment during fin sterile inflammation due to increased amounts of cortisol correlated with interrnenal tissue.

Major points:

Does cilostazol effect total neutrophil number or neutrophil biogenesis? additionally, this result could be strengthened by using a full GFP reporter (neutrophil, lyzC) besides a granulo protein mpx, which could also be done by time lapse. This could also shed light on the decrease in the initial phase of neutrophil recruitment decrease seen at 12hpi. Further more, since the claim of less neutrophil recruitment is of a retention phenotype, I would suggest a LTB4 bath model to assess neutrophil swarming to test this hypothesis.

Discuss the neutrophil specific cilostazol effect, as a cortisol induced effect usually is accompanied by less initial recruitment

qRT of pro/anti inflammatory genes to help elucidate the mechanism for this decreased neutrophil recruitment

I would recommend to tone down the discussion in terms of clinical or developmental correlations, as there is not enough evidence to support the extrapolations.

minor points:

clarify annotation for a,b,c,d,e,f ect on the figures

there is no Fig 5. corresponding text (should be in the section regarding no effect on tissue regeneration)

Reviewer #2: Review file uploaded as an attachment.

Summary:

The authors of this manuscript seek to investigate the effects of the drug Cilostazol on steroidogenesis and immune cell activity using larval zebrafish as a model. Cilostazol is a PDE3 inhibitor that generally causes an increase in cellular cAMP levels in affected tissues by preventing metabolism to AMP, which may alter physiology. The authors treat larval zebrafish with this drug and validate is efficacy by verifying it can increase zebrafish heart rate. They then provide evidence to suggest that treatment of the drug can alter steroidogenesis by increasing the number of active interrenal cells in the fish and increasing whole-body cortisol levels. The authors then investigate cilostazol-mediated changes in immune cell activity by utilizing a tail fin clip injury model. While tail fin regrowth and macrophage recruitment was not affected by drug treatment, the authors claim that neutrophil migration and retention in the injury site is altered. The authors interpret this data to suggest that cilostazol can adversely modulate sterile inflammatory processes during development and caution against treating young developing patients with the drug.

6. PLOS authors have the option to publish the peer review history of their article (what does this mean?). If published, this will include your full peer review and any attached files.

Reviewer #1: **Yes: **Alan Y. Hsu

Reviewer #2: No

---

## [Author Response · Author response to Decision Letter 0]

12 Jun 2023

On behalf of my coauthors, I would like to thank the editor and the two reviewers for taking precious time to offer critical comments and helpful advice on this article. We have addressed each comment and suggestion raised by the reviewers in a dedicated "Response to Reviewers" section following the revised manuscript.

---

## [Decision Letter · Decision Letter 1]

2 Oct 2023

The anti-platelet drug cilostazol enhances heart rate and interrenal steroidogenesis and exerts a scant effect on innate immune responses in zebrafish

PONE-D-23-00030R1

Dear Dr. Liu,

We’re pleased to inform you that your manuscript has been judged scientifically suitable for publication and will be formally accepted for publication once it meets all outstanding technical requirements.

Kind regards,

Mohammed Fouad El Basuini, Professor

Academic Editor

PLOS ONE

Additional Editor Comments (optional):

Reviewers' comments:

Reviewer's Responses to Questions

**Comments to the Author**

1. If the authors have adequately addressed your comments raised in a previous round of review and you feel that this manuscript is now acceptable for publication, you may indicate that here to bypass the “Comments to the Author” section, enter your conflict of interest statement in the “Confidential to Editor” section, and submit your "Accept" recommendation.

Reviewer #1: All comments have been addressed

Reviewer #2: All comments have been addressed

2. Is the manuscript technically sound, and do the data support the conclusions?

Reviewer #1: Yes

Reviewer #2: Yes

3. Has the statistical analysis been performed appropriately and rigorously? 

Reviewer #1: Yes

Reviewer #2: Yes

4. Have the authors made all data underlying the findings in their manuscript fully available?

Reviewer #1: Yes

Reviewer #2: Yes

5. Is the manuscript presented in an intelligible fashion and written in standard English?

Reviewer #1: Yes

Reviewer #2: Yes

6. Review Comments to the Author

Reviewer #1: I understand the difficulty and feasibility of your predicament. I would like to point out that even if something has been published it may not hold true in your system. It is ok to reference it to support your story but some validation is needed on your own. It is good practice to have rigorous validation of some results when it is a crucial link in your story besides a reference. With that said I will still endorse this story as I do "believe" it is the case.

Reviewer #2: -The authors have greatly improved on the manuscript.

-ImageJ is a part of the "Fiji" package, not "Fuji".

-lines 465 - 470. Authors are contrasting two similar sentences. Should replace "on the other hand" with something like "also" so that their upcoming contrasting sentence starting with "however" makes more sense.

7. PLOS authors have the option to publish the peer review history of their article (what does this mean?). If published, this will include your full peer review and any attached files.

Reviewer #1: **Yes: **Alan Y. Hsu

Reviewer #2: **Yes: **Logan Ganzen

---

## [Editor Report · Acceptance letter]

20 Oct 2023

PONE-D-23-00030R1 

The anti-platelet drug cilostazol enhances heart rate and interrenal steroidogenesis and exerts a scant effect on innate immune responses in zebrafish 

Dear Dr. Liu:

I'm pleased to inform you that your manuscript has been deemed suitable for publication in PLOS ONE. Congratulations! Your manuscript is now with our production department. 

Kind regards, 

on behalf of

Dr Mohammed Fouad El Basuini 

Academic Editor

PLOS ONE